# In Vivo, In Vitro and In Silico Study of *Cucurbita moschata* Flower Extract: A Promising Source of Natural Analgesic, Anti-Inflammatory, and Antibacterial Agents

**DOI:** 10.3390/molecules28186573

**Published:** 2023-09-11

**Authors:** Md. Rabiul Hossain, Rashedul Alam, Hea-Jong Chung, Taslima Akter Eva, Mohammed Fazlul Kabir, Husnum Mamurat, Seong-Tshool Hong, Md. Al Hafiz, S. M. Moazzem Hossen

**Affiliations:** 1Department of Pharmacy, University of Science and Technology, Foy’s Lake, Chittagong 4202, Bangladesh; rabiul.hossain@ustc.ac.bd (M.R.H.); husnum.mamurat@gmail.com (H.M.); 2Department of Biochemistry and Molecular Biology, Medical University of South Carolina, Charleston, SC 29425, USA; rashedcmu@gmail.com; 3Gwanju Center, Korea Basic Science Institute, Gwanju 61715, Republic of Korea; 4Department of Pharmacy, University of Chittagong, Chittagong 4331, Bangladesh; eva663300@gmail.com; 5Department of Biological Science, University at Buffalo, Buffalo, NY 14260, USA; kabirrasel07@gmail.com; 6Department of Biomedical Sciences, Institute for Medical Science, Jeonbuk National University Medical School, Jeonju 54907, Republic of Korea; seonghong@jbnu.ac.kr; 7Department of Pharmacy, East West University, Dhaka 1212, Bangladesh; alhafiz.spl@gmail.com

**Keywords:** *Cucurbita moschata*, pumpkin flower, analgesic, anti-inflammatory, antibacterial, writhing test, molecular docking, in silico study

## Abstract

For thousands of years, medicinal plants have played a pivotal role in maintaining human health and improving the quality of human life. This study was designed to analyze the analgesic, anti-inflammatory, and antibacterial potentials of a hydro-methanolic extract of *Cucurbita moschata* flowers, along with qualitative and quantitative phytochemical screening. The anti-inflammatory effect was tested using the in vitro membrane stabilizing method for human red blood cells (HRBC), the analgesic effect was tested using the in vivo acetic acid-induced writing method, and the antibacterial effect was tested using the disc diffusion method. In silico ADME/T and molecular docking studies were performed to assess the potential of the stated phytochemicals against Cyclooxygenase-II enzyme. Phytochemical screening confirmed the presence of flavonoids, alkaloids, glycosides, tannins, and carbohydrates. The flower extract demonstrated the maximum protection of human red blood cells at 1000 µg/mL, with a 65.73% reduction in hemolysis in a hypotonic solution. The extract also showed significant (*p* < 0.05) and dose-dependent analgesic effects at oral doses of 200 and 400 mg/kg on the tested animals. Furthermore, the flower extract exhibited potent antibacterial activity due to the disc diffusion method, which was compared with standard ciprofloxacin. In silico testing revealed that 42 phytochemicals exhibited notable pharmacokinetic properties and passed drug likeness screening tests. Among the six best-selected compounds, 3,4-dihydro-2H-pyran-2-yl)methanamine showed the highest binding affinity (−10.1) with significant non-bonding interactions with the target enzyme. In conclusion, the hydro-methanolic extract of *Cucurbita moschata* was found to be rich in various phytochemicals that may be associated with therapeutic potential, and this study supports the traditional use of *Cucurbita moschata* flowers in the management of inflammation and painful conditions.

## 1. Introduction

Pain is commonly understood to be an unpleasant sensory and emotional experience that can be linked to or explained in terms of actual or prospective tissue injuries [1]. It is often triggered by unpleasant stimuli and conveyed to the central nervous system (CNS) via specific neuronal networks where it is regarded intrinsically. It protects the body from harm that may otherwise come to it [2]. On the other hand, inflammation is the body’s protective response to unpleasant stimuli, such as infections and injuries, and it is mediated by both the adaptive and innate immune systems [3,4]. Pain and inflammation tend to be the most unpleasant and severe health concerns that affect 80% of individuals worldwide. This is true despite the fact that suitable drugs are available to treat both conditions [5]. Moreover, globally, pain is a clinical, social, and economic issue. It is estimated that the global cost of pain is as high as 3.0% of GDP (Gross Domestic Product), illustrating the substantial economic impact of pain. In terms of annual costs, pain is more expensive than cardiovascular illnesses or cancers are [6], and it has been shown in studies that physical pain is a reliable predictor of suicidal ideation and behavior [7].

The progression of serious inflammatory diseases, such as rheumatoid arthritis, glomerulonephritis, inflammatory bowel disease, hypersensitivities, hay fever, atherosclerosis, and asthma, is also caused by unresolved inflammation. If they are not treated and managed effectively, these debilitating conditions—which are the leading cause of disabilities—can be fatal [8].

Steroids and Nonsteroidal anti-inflammatory drugs (NSAIDs) are often used to relieve inflammation and pain. By inhibiting Cyclooxygenase (COX) enzymes, especially COX-II, NSAIDs relieve inflammation and pain. However, the prolonged use of NSAIDs has been linked to a variety of adverse liver and GIT (Gastrointestinal tract) complications. Additionally, they have effects on the kidney and the cardiovascular system [9,10]. Since NSAIDs are toxic to pathways, causing oxidative damage, researchers have highlighted the crucial role of antioxidants as natural medications for the remedy of NSAID-induced toxic effects [11,12]. This is why there has been a rise in the use of herbal drugs for the treatment of pain and inflammation. Hence, the approval of traditional South Asian therapeutic plants is gaining ground.

*Cucurbita moschata* is commonly called ‘*MistiKumra*’ in Bangla or Pumpkin in English, is an important horticultural crop that belongs to the family Cucurbitaceae [13]. The nutritional, health-protective value and presence of various phytochemicals make this plant interesting to many researchers. In the last few decades, many researchers have studied *Cucurbita moschata* and have found that *Cucurbita moschata* has many medicinal applications, such as anthelmintic, anti-diabetes, antibacterial, anticancer, and anti-obesity properties [14]. This plant has also been an integral part of rural as well as some urban diets around the world since antiquity [15]. All of its structural components are consumable, contain minimal calories, and are easily digested [16]. Phytoconstituents like alkaloids, flavonoids, palmitic, oleic, and linoleic acids are present in this plant [17]. In addition, the seeds of this plant are used in the formulation of cosmetic products, while the peel extract is reported to have burn wound healing activity, and the pulp has exhibited several activities, like antioxidant, anti-inflammatory, and anti-angiogenesis ones, along with anti-fatigue activities in mice models [18].

As there is no existing report on the analgesic, anti-inflammatory, and antibacterial effects of the hydro-methanolic extract of Bangladeshi pumpkin flowers, the aim of this study was to investigate its bioactive phytochemicals, analgesic, antibacterial, and anti-inflammatory potential through in vitro, in vivo, and in silico studies.

## 2. Results

### 2.1. Phytochemical Screening

#### 2.1.1. Qualitative Screening

A variety of phytochemicals, including alkaloids, flavonoids, phenolic compounds, reducing sugars, glycosides, and tannins, were detected in the flower extracts upon phytochemical screening, as shown in Table 1.

#### 2.1.2. Quantitative Screening

The results obtained reveal that the extract of *Cucurbita moschata* contains significant amounts of TPC and TFC. The Total Phenolic content was found 18.16 ± 0.28 mg/g gallic acid equivalent in reference to the standard curve (y = 0.001x + 0.013, R² = 0.998), and the total flavonoids content was found 10.42 ± 0.10 mg/g quercetin equivalent in reference to the standard curve (y = 0.001x + 0.045, R² = 0.997) (Table 2).

### 2.2. Anti-Inflammatory Activity

#### HRBC Membrane Stabilizing Assay

The hydro-methanolic extract of the *Cucurbita moschata* flower was investigated for in vitro anti-inflammatory activity. The extract showed maximum protection and minimum hemolysis of the HRBC, which equate to 65.73% and 34.27%, respectively, at a concentration of 1000 µg/mL in a hypotonic solution. Furthermore, at 125 µg/mL, the extract exhibited maximum hemolysis and minimum protection values of 80.35% and 19.65%, respectively. The results were compared with the standard drug diclofenac, which showed a value of 90.89% for protection and a value of 9.11% for hemolysis (Figure 1).

### 2.3. Analgesic Activity

#### Acetic Acid-Induced Writhing Test

Table 3 shows the effects of the hydro-methanolic extract of *Cucurbita moschata* flower on acetic acid-induced writhing among experimental mice. The extract exhibited a significant reduction (*p* < 0.05) of writhing induced by acetic acid after intraperitoneal administration in a dose-dependent manner. After the oral administration of two different doses, 200 and 400 mg/kg body weight, the percent inhibition was found to be 25.53% and 56.03%, respectively. Standard drug diclofenac sodium (80.85%) was found to be more potent than the plant extracts were.

### 2.4. Antibacterial Activity

The test agents’ effectiveness as antibacterial substances was evaluated based on their ability to limit bacterial growth around the discs, creating a clear zone of inhibition. Following incubation, the diameter of the zone of inhibition resulting from using the test materials was measured in millimeters using a transparent scale. This provided information on their antimicrobial efficacy, which is recorded in Table 4 and Figure 2.

The *Cucurbita moschata* flower test sample exhibited antibacterial activity, with a 31 mm inhibition zone observed against *E. coli* and 29 mm and 21.67 mm inhibition zones against *Staphylococcus aureus* and *Pseudomonas aeruginosa*, respectively. Based on the data presented, it can be concluded that the effectiveness of the extract and the standard antimicrobial agent (Ciprofloxacin) varied depending on the bacterial strain tested, where the *Cucurbita moschata* extract was more effective against *Staphylococcus aureus* and *E. coli* than the standard antibiotic was.

### 2.5. In Silico Study 

#### 2.5.1. ADME/T and Drug-Likeness Analysis

In order to demonstrate their potential as therapeutic targets, reported phytochemicals were screened for their pharmacokinetic and drug likeliness qualities prior to docking analysis. The pharmacokinetic profiles of the reported phytochemicals are shown in Table 5, which depict that they do not violate any of the criteria for the Lipinski rule and have no mutagenic and carcinogenic properties. In silico ADMET and drug likeness studies of the compounds present in *Cucurbita moschata* using pKCSM and SwissADME online tools showed notable results. Of the eighty-four compounds examined, forty-two passed the Lipinski’s rule of five for drug likeness without exhibiting any toxicity. The other compounds which do not follow the Lipinski’s properties and exhibit a poor pharmacokinetic profile with organ toxicity were not considered for further docking studies.

#### 2.5.2. Molecular Docking and Post-Docking Analysis

A docking score with non-bonding interactions was used as the parameter for the binding interactions and affinity of the compounds, along with those of standard drugs, Diclofenac and Celecoxib (Table 6). Of the total of forty-two selected compounds, 3,4-dihydro-2H-pyran-2-yl)methanamine (−10.1 kcal/mol) showed the highest binding affinity, which was higher than those of the standard drugs Celecoxib (−9.5 kcal/mol) and Diclofenac (−8.0 kcal/mol). The docking scores of five compounds, i.e., 2,3-diphenyl-1H-pyrrolo[2,3-b]pyridin-6-amine, Oxacyclotridecan-2-one (−9.1), Luteolin (−8.7); (E)-1,3,7-trimethyl-8-(2-nitrostyryl)-3,7-dihydro-1H-purine-2,6-dione (−8.6), and Kaempferol, (−8.4) also showed a promising binding affinity with Cyclooxygenase II enzyme.

Using the Biovia Discovery Studio Visualizer program, the binding interactions between the best-docked molecule and the amino acid residues of the target enzyme shown in Figure 3A–F were investigated further. This analysis showed that 2H-pyran, 3,4-dihydro-2-aminomethyl had a similar binding mode and interaction with COX-2 protein-like diclofenac (Table 5, Figure 2A). The critical evaluation of the docking process showed that 2H-Pyran, 3,4-dihydro-2-aminomethyl formed eight Van der Waals bonds with GLY A:15, ASN A:18, GLN A:19, THR A:46, SER A:49, GLY A:93, GLY A:94, and PHE A:98 and four hydrophobic interactions with LEU A:28, LEU A:54, VAL A:31, and PHE A:92, with short intermolecular distances, which is suggestive that it has a good binding affinity toward the active site of COX-2 enzyme. As hydrophobic interactions are one of the major driving forces of drug-receptor interactions, these bonds with shorter distances (<5 Å) cause strong binds; hence, they have a higher docking score.

## 3. Discussion

In recent years, natural medicines made from plants have gained popularity for their ability to effectively treat pain and inflammation [19]. In Bangladeshi traditional medicine, various natural substances and formulations have been used to alleviate pain and inflammation [20,21]. In the current study, a hydro-methanolic extract of *Cucurbita moschata* flowers exhibited a potential anti-inflammatory action during membrane stabilization in an HRBC test in a dose-dependent manner (Figure 1). The extract demonstrated membrane stabilization by inhibiting the hypotonicity-induced lysis of the RBC membrane [21,22], suggesting that the extract could stabilize lysosomal membranes. During an inflammatory reaction, lysosomal components of active neutrophils, such as bactericidal enzymes and proteases, are released, which can damage the tissue further. Therefore, lysosomal stability is crucial in limiting the inflammatory response [23,24]. The phytochemicals present in the *Cucurbita moschata* flower extract may inhibit these processes and improve the efflux of intracellular components [25].

The acetic acid-induced abdominal contraction response can be effectively studied to detect the peripheral analgesic impact, with specific peritoneal receptors playing a role in the abdominal contraction mechanism [26,27]. The acetic acid-induced writhing method draws out a pain sensation by activating a local inflammatory reaction, which is a consequence of free arachidonic acid being released from tissue phospholipids [28,29]. This reaction leads to an increase in the levels of prostaglandin E2 (PGE2) and prostaglandin F2α (PGF2α) in the peritoneal fluids, as well as lipoxygenase products [30]. The elevated prostaglandin level in the peritoneal cavity causes an increment of pain by expanding the capillary permeability [31].

An agent that reduces the amount of writhing among acetic acid-induced mice provides an analgesic effect, preferably by inhibiting prostaglandin synthesis, a peripheral mechanism of pain inhibition [29,32]. The significant pain reduction observed with the *Cucurbita moschata* flower extract may be due to the presence of analgesic principles acting on the prostaglandin pathways.

Again, various qualitative phytochemical screening tests can be used to identify the presence of phenols, flavonoids, alkaloids, glycosides, tannins, and carbohydrates. In quantitative phytochemical screening, significant amounts of phenolic and flavonoid contents were identified (Table 2). The analgesic and anti-inflammatory effects of plant-derived phytochemicals like phenols, flavonoids, alkaloids, glycosides, tannins, and carbohydrates have been reported [33].

The extract of the flower exhibited an antibacterial effect in tests conducted on six different strains of bacteria. The efficacy of the test agents in terms of their antibacterial properties were evaluated based on their ability to impede the growth of bacteria surrounding the discs, resulting in a clear zone of inhibition. The presence of phenols, flavonoids, alkaloids, glycosides, and tannins in the extract may account for its antibacterial effect, as these phytochemicals are known to possess antibacterial properties [34,35,36,37,38,39,40].

In silico ADME/T and molecular docking studies are a good way to come up with new drugs for a wide range of diseases. This technique has the advantage of consuming less time and requiring less money than traditional lab experiments do [41]. The phytochemicals reported in *Cucurbita moschata* [42,43] were evaluated for pharmacokinetic parameters and drug likeliness, and most of the compounds showed notable results (Table 5). Among them, forty-two compounds passed the drug-likeness study with no toxicity. In this research, we used a docking method making use of open software programs as well as virtualized interactions of the ligands 3,4-dihydro-2H-pyran-2-yl)methanamine, Oxacyclotridecan-2-one, Luteolin, and Kaempferol with COX-II enzyme. An important phase in ligand docking in favorable conformations is the formation of non-bonding interactions, including hydrogen bonds, hydrophobic interactions, and Van der Waals interactions with essential amino acids, as revealed by the docking score. According to our findings, each ligand–protein interaction is mediated by a large amount of amino acid residues in the hydrophobic and Van der Waals bond interactions. 3,4-dihydro-2H-pyran-2-yl)methanamine showed the highest docking score, which was higher than the standard drugs, Celecoxib and Diclofenac, with excellent non-bonding interactions with Cyclooxygenase-II enzyme (Figure 2). Additionally, it should be noted that this substance has excellent pharmacokinetic characteristics (Table 5) and also passed the drug-likeness screening test. These observations further confirm that 3,4-dihydro-2H-pyran-2-yl)methanamine may be a more effective anti-inflammatory and analgesic compound, especially with respect to COX-2 protein-mediated inflammation and pain, compared to other traditional NSAIDs.

## 4. Materials and Methods

### 4.1. Plant Material

Pumpkin flowers were obtained locally in Chattogram City, Bangladesh, and were authenticated by Dr. Shaikh Bokhtear Uddin, Professor, Department of Botany, University of Chittagong. A specified amount was stored at the Department of Pharmacy, University of Science and Technology Chittagong (USTC).

### 4.2. Preparation of Plant Extract

The flowers were cleaned properly with running tap water and dried in sunlight after being collected and identified. The flowers were finely powdered using a blender after drying. About 1000 g *Cucurbita moschata* flowers powder was then extracted with 80% methanol in water for 72 h while continuously stirring with a magnetic stirrer at room temperature. Following that, the flower extract was filtered through a clean cotton filter, and then through Whitman filter paper. The solvent was evaporated with a Rotary evaporator (Lab Tech EV311) at 40 °C under reduced pressure. The extract was then refrigerated (2–8 °C) until further usage.

### 4.3. Experimental Animals

Swiss albino mice of either sexes weighing 20–30 g were obtained from Animal Research Branches of BCSIR, Chittagong. In the animal house of the Department of Pharmacy, USTC, they were housed in dry and clean iron cages under a 12 h light and dark cycle at 25 ± 3 °C and 45–55% relative humidity. The mice were given free access to water and a typical laboratory diet supplied by the BCSIR laboratory. Food was withdrawn 12 h before the experiment and while it was going on. Given their acute sensitivity to environmental changes, these animals were acclimated to the laboratory for a full week before the trial began. All experiments were conducted in a quiet, soundproofed space. The investigation was permitted by the institutional ethics committee of USTC under ethical approval number USTMEBBC/20/02/05.

### 4.4. Phytochemical Screening

#### 4.4.1. Qualitative Screening

The presence of different phytochemicals, such as an alkaloid, carbohydrate, flavonoid, glycoside, tannin, steroid, and saponin, was qualitatively assessed in the freshly prepared crude extract. Standard protocols were used to identify them based on distinctive color changes [44].

#### 4.4.2. Quantitative Screening

##### Total Phenolic Content

The Folin–Ciocalteu technique was used to determine the Total Phenolic content of the extract [45]. Briefly, 200 µL of the crude extract (1 mg/mL) was diluted to 3 mL with distilled water, mixed well with 0.5 mL of Folin–Ciocalteu reagent (10%) for 3 min, and then 2 mL of 20% (*w*/*v*) sodium carbonate was added. After another 60 min in the dark, the mixture was checked for absorbance at 650 nm. The calibration curve was used to figure out the Total Phenolic content, which was written as mg of gallic acid equivalent per gram of dry weight.

##### Total Flavonoid Content

The aluminum chloride colorimetric technique was used to determine the total flavonoid content of crude extract [46]. Briefly, 50 µL of crude extract (1 mg/mL) was converted into 1 mL with methanol and mixed with 4 mL of distilled water and 0.3 mL of 5% NaNO_2_ solution; 0.3% of 10% AlCl_3_ solution was added after 5 min of incubation, and the mixture was allowed to stand for 6 min. Then, 2 mL of 1 mol/L NaOH solution was added, and the final volume of the mixture was brought to 10 mL with double distilled water. After allowing the mixture to sit for 5 min, the absorbance was measured at 510 nm. The total flavonoid content was determined using a calibration curve and is expressed as mg of quercetin per gram of dry weight.

### 4.5. Anti-Inflammatory Study

#### Human RBC Membrane Stabilization Assay

Human red blood cells membrane stabilization was used as a method to evaluate the anti-inflammatory activity [47]. The collected blood was mixed with an equal volume of sterilized Alsever medium (2%, (*w*/*v*) dextrose, 0.8% sodium citrate, 0.5% citric acid, and 0.42% sodium chloride in water). The blood was further centrifuged at 3000 rpm for 10 min, and the packed cells were washed with isosaline (0.85%, pH 7.2), and finally, 10% (*v*/*v*) suspension was made with isosaline. The assay mixture contained the extract, 1 mL phosphate buffer (0.15 M, pH 7.4), 2 mL hyposaline (0.36%), and 0.5 mL human RBC suspension. Diclofenac sodium was used as a standard drug. Instead of hyposaline, 2 mL distilled water was used as the control. The assay mixtures were incubated at 37 °C for 30 min and centrifuged at 3000 rpm for 10 min. The hemoglobin content in the supernatant was estimated using a UV–Visible spectrophotometer at 560 nm. The percentage protection was calculated using the following equation:

The percentage of hemolysis of HRBC membrane was calculated as follows: $$\%\;Hemolysis=\frac{Optical\;density\;of\;test\;sample}{Optical\;density\;of\;control}\;\times \;100$$

The percentage of HRBC membrane stabilization (protection) was calculated as follows: $$\%\;protection=100-(\frac{Optical\;density\;of\;test\;sample}{Optical\;density\;of\;control}\;\times \;100)$$

### 4.6. Analgesic Activity

#### Acetic Acid-Induced Writhing Test

This test was performed according to Zulfiker AH et al. [29]. The mice were randomly divided into 4 groups designed as group I, group II group III, and group IV, each containing 6 individuals for the control, diclofenac sodium, and extract treated groups, respectively. The analgesic effect of the hydro-methanolic extract of *Cucurbita moschata* was investigated by giving 200 and 400 mg/kg (10 mL/kg body weight) extract via oral administration to mice. Diclofenac (50 mg/kg; 10 mL/kg body weight) was used as a positive control, and it was dissolved in 1% tween 80. For the negative control animals, 10 mL/kg body weight of water was administered using the same protocol. Each mouse was injected intra-peritoneally with 0.7% (*v*/*v*) acetic acid at a dose of 10 mL/kg body weight after 30 min of each respective treatment. After 15 min of the acetic acid treatment, the number of writhing reactions by each animal was recorded over the course of 5 min. During writhing motions, the muscles in the abdomen region contract, while the muscles in the hind limbs stretch. The percentage of inhibition was calculated using the following ratio:
$$\%\;\mathrm{inhibition}=(\frac{\mathrm{Control}\;\mathrm{mean}-\mathrm{treated}\;\mathrm{mean}}{\mathrm{Control}\;\mathrm{mean}})\;\times \;100$$

### 4.7. Antibacterial Activity

#### Disc Diffusion Method

The conventional method employed in this study involves creating a concentration gradient through the diffusion of antibiotics from a contained source into nutrient agar gel [48]. As a positive and negative control, standard antibiotic (Ciprofloxacin 5 µg/mL) discs and blank discs are used, respectively. Test samples, in predetermined quantities, are placed on sterile and dried nutrient agar discs (6 mm diameter), which are then inoculated with the test microorganisms. This results in the formation of a zone of inhibition when the discs are exposed to compounds that restrict microbial growth in the surrounding media. For this study, pure cultures of six bacterial strains, namely *Staphylococcus aureus*, *Salmonella typhi*, *Proteus vulgaris*, *E. coli*, *Klebsiella pneumoniae*, and *Pseudomonas aeruginosa*, were obtained from the Department of Microbiology, University of Chittagong.

To ensure accuracy, antimicrobial screening was performed in a laminar flow hood with strict adherence to all necessary safety measures to eliminate the possibility of contamination and cross-contamination with the test organisms. Prior to entering the laminar hood, UV lighting was activated for an hour. Autoclaving at 121 degrees Celsius and 15 pounds per square inch pressure for 20 min are the recommended conditions for sterilizing glassware, such as Petri dishes, cotton, forceps, micropipette tips, and blank discs.

To perform the test, discs were soaked in a solution of 50 μL drawn from a 1 mg/mL sample solution and carefully placed in the designated areas of the agar plates that were inoculated with test bacteria. The plates were then inverted and stored in a 40 °C refrigerator for approximately 24 h to facilitate diffusion of the components from the discs to the surrounding agar medium. Subsequently, the plates were inverted and incubated for 24 h at 37 °C. The diameter of the zone of inhibition produced by the flower extract was then determined and compared to that produced by standard antibiotic ciprofloxacin (5 µg/mL).

### 4.8. In Silico Study

#### 4.8.1. Selection of Ligands

A total of eighty-two compounds were retrieved after a thorough review of the literature and our previously completed GCMS analyses. Compounds, namely, 3,4-dihydro-2H-pyran-2-yl)methanamine-,2,3-diphenyl-1H-pyrrolo[2,3-b]pyridin-6-amine,Oxacyclotridecan-2-one, Luteolin, (E)-1,3,7-trimethyl-8-(2-nitrostyryl)-3,7-dihydro-1H-purine-2,6-dione, Kaempferol, etc. [42,43], were retrieved from the PubChem database (pubchem.ncbi.nlm.nih.gov).

#### 4.8.2. Validation of the Ligands as Potential Therapeutic Agents

The physical, molecular features and pharmacokinetic parameters like ADME/T (absorption, distribution, metabolism, excretion, and toxicity) of compounds play a vital role in the selection of these agents as drug candidates. These properties of the listed compounds were analyzed by using the pKCSM online tool (http://biosig.unimelb.edu.au/pkcsm/) (accessed on 31 July 2021) to validate them as potential ligands against therapeutic targets [49]. The compounds were then filtered through Lipinski’s rule of five to predict their drug likeliness using the SwissADME online server [50]. Forty-two out of eighty-four compounds were suitable for further docking analysis, and forty-two compounds were rejected due to violation of Lipinski’s rule for more than one parameter.

#### 4.8.3. Protein Preparation and Active Site Determination

The crystal structure of target protein COX2 with a selective inhibitor with a resolution of 2.80 Å (PDB ID: 6COX) was collected from the RCSB protein data bank [51]. The active site of the enzyme was identified using previously given information from Kurumbail et al. [52]. Necessary cleaning and preparations, like the deletion of heteroatoms, cofactor, and water, were achieved using BIOVIA Discovery Studio4.5 Client and Swiss-PdbViewer (v4.1). Hydrogen atoms were added to their geometry, and the target protein was minimized using MMFF94s force field with the PyRx-virtual screening tool. The target protein was saved in pdb format for docking investigations.

#### 4.8.4. Molecular Docking and Post-Docking Analysis

Docking calculations were performed with AutoDock, version 4.2, using the PyRx 0.3 (http://pyrx.scripps.edu) (accessed on 31 July 2021) [53]. A grid box size of X: 46.0156; Y: 48.1357; Z: 43.0279 Å points with a grid spacing of 0.375 Å was generated using AutoGrid. Analysis of the docking results was performed using PyMOL. Such tools can help elucidate which type of interaction (e.g., hydrogen bond, π-π interaction, and cation-π interaction) contribute to ligand binding. PyMOL was used to provide complementary information on ligand–receptor interaction [54].

### 4.9. Statistical Analysis

All the results are expressed as mean ± standard error of mean (SEM). All the data were analyzed statistically by One-way ANOVA, followed by Dunnett’s *t*-test. The values were obtained were compared with the vehicle control group and were considered statistically significant when *p* < 0.05.

## 5. Conclusions

The hydro-methanolic extract of *Cucurbita moschata* (Pumpkin) flower has shown great promise in the area of natural medicine, according to the results of our study. We have found that this extract is rich in bioactive phytochemicals that exhibit anti-inflammatory, antibacterial, and analgesic properties both in vitro and in animal studies.

One compound of particular interest that we have identified in our experiment is 3,4-dihydro-2H-pyran-2-yl)methanamine, which may have the potential to serve as a selective COX-2 inhibitor for the treatment of pain and inflammation. However, further detailed studies are needed to determine the efficacy of this compound as an anti-inflammatory and analgesic molecule.

In conclusion, the hydro-methanolic extract of *Cucurbita moschata* (Pumpkin) flower has shown significant promise as a natural remedy for various health conditions. The bioactive phytochemicals found in this extract demonstrate anti-inflammatory, antibacterial, and analgesic properties that could potentially be harnessed to create effective treatments for a range of ailments. Future research is necessary to fully understand the therapeutic potential of these compounds and to develop safe and effective treatment options for patients.

## Figures and Tables

**Figure 1 molecules-28-06573-f001:**
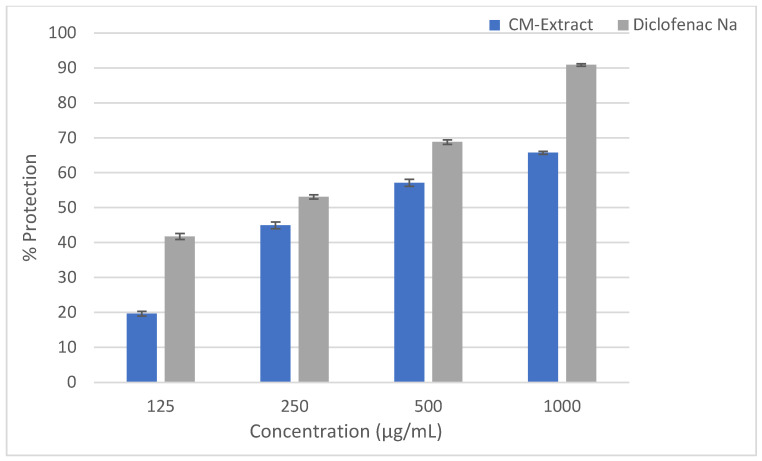
Effect of the hydro-methanolic extract of *Cucurbita moschata* flower on percent protection in HRBC membrane stabilizing method. Data are represented as mean ± SEM (*n* = 3).

**Figure 2 molecules-28-06573-f002:**
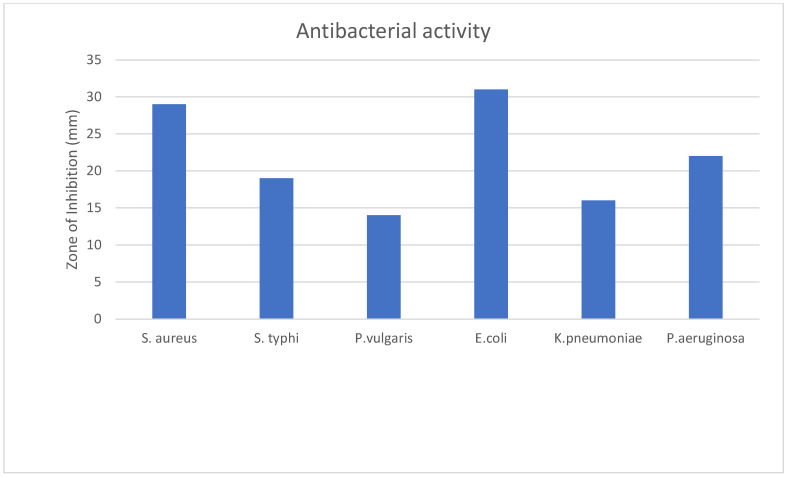
Antibacterial activity of *Cucurbita moschata* flower against six bacterial strains.

**Figure 3 molecules-28-06573-f003:**
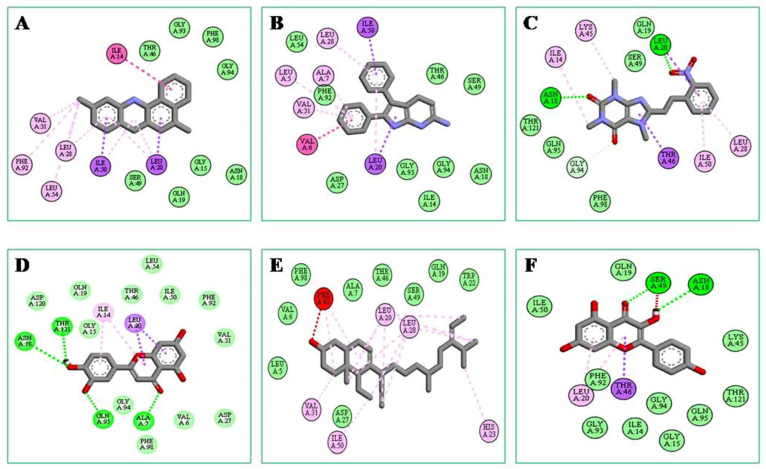
In silico non-bonding interaction of six best selected ligands with Cyclooxygenase-II enzyme ((**A**) 2H-Pyran, 3,4-dihydro-2-aminomethyl- (−10.1); (**B**) 1H-Pyrrolo[2,3-B]pyridin-6-amine, 2,3-diphenyl- (−10), (**C**) Oxacyclotridecan-2-one (−9.1); (**D**) Luteolin (−8.7); (**E**) Purin-2,6-dione, 1,3,9-trimethyl-8-[2-nitrophenethenyl] (−8.6); (**F**) Kaempferol (−8.4)).

**Table 1 molecules-28-06573-t001:** Qualitative phytochemical analysis of hydro-methanolic flower extract of *Cucurbita moschata*.

Phytochemical Constituents	Specific Tests	Inference
Alkaloids	Mayer’s test	**+**
	Hager’s test	**+**
	Wagner test	**-**
Carbohydrates	Molisch’s test	**+**
	Benedict’s test	**-**
	Fehling’s test	**+**
Flavanoids	Alkaline reagent test	**+**
Phenols	Ferric chloride test	**+**
Saponins	Foam test	**-**
Tannins	Gelatin test	**+**
Glycosides	Liebermann’s test	**+**

+: Presence; -: absence.

**Table 2 molecules-28-06573-t002:** TPC and TFC of hydro-methanolic flower extract of *Cucurbita moschata*.

Extracts	Total Phenolic Content (TPC) mg/g GAE	Total Flavonoids Content (TFC) mg/g GAE
CM extract	18.16 ± 0.28	10.42 ± 0.10

Note: Data are mean ± SEM (*n* = 3).

**Table 3 molecules-28-06573-t003:** Effect of the hydro-methanolic extract of *Cucurbita moschata* flower on acetic acid-induced writhing among mice.

Groups	Treatment	Dose &Route	No. of Writhing	% Inhibition
G-I	1% Tween 80 (Control)	10 mL/kg; p.o	23.5 ± 0.91	NA
G-II	Diclofenac Na (Standard)	50 mg/kg; p.o	4.5 ± 0.61	80.85
G-III	CM extract	200 mg/kg; p.o	17.5 ± 0.84 *	25.53
G-IV	CM extract	400 mg/kg; p.o	10.33 ± 0.56 *	56.03

Each values represent the mean ± SEM of 6 experiments; * *p* < 0.05 Dunnett’s *t*-test as compared to control.

**Table 4 molecules-28-06573-t004:** Antibacterial activity of *Cucurbita moschata* flower against pathogenic bacterial strains.

Sample	*Stephylococcus aureus*	*Salmonella 88typhi*	*Proteus vulgaris*	*E. coli*	*Klebsiella pneumoniae*	*Pseudomonas aeruginosa*
	*Zone of Inhibition* (mm)					
CM Extract (1 mg/mL)	29 ± 0.58	19.33 ± 0.88	14 ± 1.15	31 ± 1.16	16.33 ± 0.33	21.67 ± 1.45
Ciprofloxacin (5 µg/mL)	21.33 ± 0.	31.67 ± 0.66	19.67 ± 1.76	25.33 ± 0.33	22.33 ± 0.89	23.33 ± 0.88

**Table 5 molecules-28-06573-t005:** In silico ADME/T and drug likeliness study of reported phytochemicals of *Cucurbita moschata* flower.

Compounds Name	Absorption		Distribution		Metabolism	Excretion	Toxicity		Drug Likeliness	Bioavailability
	Water Solubility (log mol/L)	Intestinal Absorption (Human) (% Absorbed)	VDss (Human) (log L/kg)	BBB Permeability(log BB)	CYP3A4 Substrate	Total Clearance (log ml/min/kg)	AMES Toxicity	Hepato Toxicity		
Protocatechuic acid	−2.07	71.17	−1.29	−0.683	No	0.551	No	No	Yes	0.56
Vanillic acid	−1.84	78.15	−1.74	−0.38	No	0.719	No	No	Yes	0.85
Caffeic acid	−2.33	69.41	−1.09	−0.647	No	0.508	No	No	Yes	0.56
Syringic acid	−2.22	73.07	−1.44	−0.191	No	0.646	No	No	Yes	0.56
Ferulic acid	−2.82	93.68	−1.36	−0.239	No	0.623	No	No	Yes	0.85
trans-sinapic acid	−2.87	93.06	−1.11	−0.247	No	0.718	No	No	Yes	0.85
Tyrosol	−1.15	85.26	−0.11	−0.218	No	0.283	No	No	Yes	0.56
Luteolin	−3.09	81.13	1.15	−0.907	No	0.495	No	No	Yes	0.55
Kaempferol	−3.04	74.29	1.274	−0.939	No	0.477	No	No	Yes	0.56
3,4-dihydro-2H-pyran-2-yl)methanamine	−0.06	96.57	0.27	−0.229	No	1.038	No	No	Yes	0.56
5,10-dimethylbenzo[c]acridine-	−5.61	99.45	0.13	0.645	Yes	0.851	Yes	Yes	Yes	0.56
2,3-diphenyl-1H-pyrrolo [2,3-b]pyridin-6-amine	−3.41	94.96	−0.463	0.464	Yes	0.542	Yes	Yes	Yes	0.85
Oxacyclotridecan-2-one	−2.759	95.30	0.142	0.408	No	1.345	No	No	Yes	0.56
(E)-1,3,7-trimethyl-8-(2-nitrostyryl)-3,7-dihydro-1H-purine-2,6-dione	−3.209	83.45	0.073	−1.362	Yes	0.045	Yes	Yes	Yes	0.55
Avenasterol	−6.715	94.64	0.179	0.764	Yes	0.613	No	No	Yes	0.55

**Table 6 molecules-28-06573-t006:** In silico binding affinity and non-bonding interaction of selected phytochemicals of *Cucurbita moschata* flowers.

Compounds Name	Docking Score	Non-Bonding Interaction		
		Hydrogen Bond	Hydrophobic Bond	Van Der Waals
3,4-dihydro-2H-pyran-2-yl)methanamine-	−10.1	ALA A:7, ASN A:18	LEU A:28, LEU A:54, VAL A:31, PHE A:92	GLY A:15, ASN A:18, GLN A: 19, THR A:46, SER A:49, GLY A:93, GLY A:94, PHE A:98
2,3-diphenyl-1H-pyrrolo[2,3-b]pyridin-6-amine	−10	THR A:21, VAL A:31	LEU A:5, ALA A:7, LEU A:28, VAL A:31	ILE A: 14, ASN A:18, ASP A:27, THR A:46, SER A: 49, LEU A:54, PHE A:92, GLY A:93, GLY A:94
Oxacyclotridecan-2-one	−9.1	ASN A:18, LEU A:20	ILE A:14, LEU A:28, LYS A:45, ILE A:50	GLN A:19, SER A:49, GLN A:95, PHE A:98, THR A:121
Luteolin	−8.7	ALA A:7, ASN A:18, GLN A:95, THR A:21	ILE A:14, LEU A:20	VAL A:6, GLY A:15, GLN A:19, ASP A:27, VAL A:31, THR A:46, ILE A:50, LEU A:54, PHE A:92, GLY A:94, PHE A:98, ASP A:120
(E)-1,3,7-trimethyl-8-(2-nitrostyryl)-3,7-dihydro-1H-purine-2,6-dione	−8.6		LEU A:20, LEU A:28, HIS A:23, VAL A:31, ILE A:50	LEU A:5, VAL A:6, ALA A:7, GLN A:19, TRP A:22, ASP A:27, THR A:46, SER A:49, PHE A:98
Kaempferol	−8.4	SER A:49, ASN A:18	LEU A:20, THR A:46	ILE A:14, GLY A:15, GLN A:19, LYS A:45, ILE A:50, PHE A:92, GLY A:93, GLY A:94, GLN A:95, THR A:121
Avenasterol	−7.8	THR A:46	LEU A:20, VAL A:31, LYS A:45, ILE A:50, PHE A:92	LEU A:5, VAL A:6, ALA A:7, ILE A:14, GLY A:15, ASP A:27, LEU A:28, GLY A:93, GLY A:94, GLN S:95, THR A:96, THR A:121
Diclofenac	−8.0	ALA A:7, ILE A:50, VAL A:31	LEU A:28, GLY A:93, PHE A:98, GLY A:15	VAL A:6, PHE A:92, GLY A:94
Celecoxib	−9.5	ARG A:89, ARG A:482, SER A:322, GLN A:161	VAL A:492, ALA A:496, GLY A:495	, LEU A:321, TYR A:354, VAL A:318

## Data Availability

Upon reasonable request, the corresponding author will provide the data that back up the study’s conclusions.

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
