# Peer review of "In Vivo, In Vitro and In Silico Study of Cucurbita moschata Flower Extract: A Promising Source of Natural Analgesic, Anti-Inflammatory, and Antibacterial Agents"

_molecules, 2023, doi:10.3390/molecules28186573_

Round 1

Reviewer 1 Report

After reading the article "In vivo, in vitro and in silico study of Cucurbita moschata Flower Extract: A Promising Source of Natural Analgesic, Anti-inflammatory, and Antibacterial Agents".

I found the results obtained by the authors interesting, however, I have some doubts:

1. What does GDP and GIT mean?

2. Table 5 is not well appreciated

3. It remains to be discussed at the molecular level how these active compounds can generate analgesic, anti-inflammatory and antibacterial effects.

4. What was the extract and diclofenac dissolved in for administration?

5. Remove sections that were not used before references.

Minor editing of English language required

Reviewer 2 Report

The authors of the aimed to  analyze the analgesic, anti-inflammatory and antibacterial potential of a hydro-methanolic extract of Cucurbita moschata flowers, along with qualitative and quantitative phytochemical screening. It can be published after following corrections:

 1-The authors must add these sentences in line 70 to emphasize importance of work "Researchers have emphasized the significance of anti-oxidants as natural medicines for the treatment of NSAIDS-induced toxicities since NSAIDS created toxicities through pathways brought on by oxidative damage." (1-2)

https://doi.org/10.1007/s11033-022-07928-7

  • https://doi.org/10.1080/01635581.2020.1801775

2- The language of the MS must be improved.

3- R square in line 89 and 90 the r must be written as capital R

4-Legends of figure 2 needs more explanation. There is no statistical analysis for this antimicrobial study. Please include it. If you have not done statistics explain your reasons

5-Please include eth,cal approval number in line 283

The language of the MS must be improved in some places. There are some typographical errors as well. 

Reviewer 3 Report

After reading the manuscript "In vivo, in vitro and in silico study of Cucurbita moschata Flower Extract: A Promising Source of Natural Analgesic, Anti-inflammatory, and Antibacterial Agents", I consider that should be significantly improved before to be considered for further revisions.

·         The introduction should be enriched with more information about Cucurbita moschatais

·         HRBC membrane stabilizing assay, a negative control should be included in the graphic to have some conclusions. Additional experimental studies should performed to be shure that the extract and components possess anti-inflammatory effect.

·         About the in silico studies, how you can assure that compounds described are present in your methanolic extract. Characterization of the main components of the extract should be characterized by experimental methods.

·         Tables are incomplete and the nomenclature is wrong.

·         Docking study is not validated.

·         The conclusions are no supported by the results.

Round 2

Reviewer 1 Report

The authors made the proposed suggestions

Minor editing of English language required

Author Response

please have a look over the attached file..

Reviewer 3 Report

After reviewing the new version, I have the following commentaries:

I insist about the importance of employing the correct nomenclature, a free tool to give the name is not enough without the correct interpretation. As an example, I suppose that “1H-Pyrrolo[2,3 B]pyri-din-6 amine, 2,3-diphe-nyl-“ should be “2,3-diphenyl-1H-pyrrolo[2,3-b]pyridin-6-amine” according to the structure in panel B at figure 3. I recommend using ChemDraw or ChemSketch to assign more accurate names.

Some observations are mentioned below, however, all the manuscript should be carefully review:

Line 39: “2H-Pyran, 3,4-dihydro-2-aminomethyl” don’t have any sense, the nomenclature is wrong.

Table 5: All the following names are incorrect and doesn’t have any sense:

·         2H-Pyran, 3,4 dihydro-2 aminomethyl-

·         Benz[c]acridine, 5,10-dimethyl-

·         1H-Pyrrolo[2,3 B]pyri-din-6 amine, 2,3-diphe-nyl-

·         Purin-2,6-dione, 1,3,9-trimethyl-8-[2-nitro-phenethenyl]-

These same names are incorrect in Table 6

Line 171 the name 2H-Pyran, 3,4-dihydro-2-aminomethyl- is wrong again.

Line 174 1H-pyrrolo[2,3-B]pyridin-6-amine, 2,3-diphenyl- is wrong again.

Line 175 Purin-2,6-dione; 1,3,9-trimethyl-8-[2-nitrophenethenyl] is wrong again.

Line 183 2H-pyran, 3,4-dihy- dro-2-aminomethyl is wrong again.

Figure 3. The name “2H-Pyran, 3,4 dihydro-2 aminomethyl-“ is not even close to the structure observed at the panel A; The same observation is true for panel C and E.

On the other hand, some additional commentaries are:

Lines 55 y 56: Pain and inflammation are not a disease; they are the consequence of a disease or injury.

Line 122: “(Figure-“ which figure?, 1?

Usually, a validation of docking study can be made by redocking the crystallographic ligand and calculating the RMSD from the starting crystallographyc coordinates of the ligand versus the docking result, RMSD should be lower than 2 Å to consider that your docking calculations are performed successfully (J Chem Inf Model. 2009 Feb; 49(2): 444–460. doi: 10.1021/ci800293n).

Author Response

please look over the attached file
